# Continuous Meta-Learning without Tasks

## Abstract

Meta-learning is a promising strategy for learning to efficiently learn within new tasks, using data gathered from a distribution of tasks. However, the meta-learning literature thus far has focused on the *task segmented* setting, where at train-time, offline data is assumed to be split according to the underlying task, and at test-time, the algorithms are optimized to learn in a single task. In this work, we enable the application of generic meta-learning algorithms to settings where this task segmentation is unavailable, such as continual online learning with a time-varying task. We present meta-learning via online changepoint analysis (MOCA), an approach which augments a meta-learning algorithm with a differentiable Bayesian changepoint detection scheme. The framework allows both training and testing directly on time series data without segmenting it into discrete tasks. We demonstrate the utility of this approach on a nonlinear meta-regression benchmark as well as two meta-image-classification benchmarks.

## 1 Introduction

Meta-learning methods have recently shown promise as an effective strategy for enabling efficient few-shot learning in complex domains from image classification to nonlinear regression (Finn et al., 2017; Snell et al., 2017). These methods leverage an offline meta-training phase, in which they use data from a distribution of tasks to optimize learning performance on new tasks. These algorithms have focused on settings with *task segmentation*, where the learning agent knows when tasks change. At meta-train time, these algorithms assume access to a meta-dataset of datasets from individual tasks, and at meta-test time, the learner is evaluated on a single task. However, there are many applications where task segmentation is unavailable, which have thus far been under-addressed in the meta-learning literature. For example, consider a robot which must learn to adapt to a changing environment. The robot may switch from one environment to another during the course of deployment, and these task switches may not be directly observed. Furthermore, using an existing time series from interaction to craft a meta-dataset may require a difficult or expensive process of detecting switches in task.

In this work, we aim to enable meta-learning in task-unsegmented settings, operating directly on time series in which the latent task undergoes discrete, unobserved switches, rather than requiring a pre-segmented meta-dataset. Equivalently, this problem can be viewed from the perspective of continual learning, in that we apply the meta-learning approach to the standard online learning problem statement wherein an agent must sequentially make predictions and learn with a potentially varying latent data generating process. To accomplish this, we integrate a Bayesian changepoint estimation scheme with existing meta-learning approaches, allowing the algorithm to reason about whether or not the task has changed in a time series. Thus, we enable a standard meta-learning algorithm, which is designed for the task segmented setting, to be both trained and tested directly on time series data without the need for task segmentation.

**Contributions.**   The primary contribution of this work is an algorithmic framework for task unsegmented meta-learning which we refer to as meta-learning via online changepoint analysis (MOCA). MOCA wraps arbitrary meta-learning algorithms in a differentiable changepoint estimation algorithm, enabling application of meta-learning algorithms directly to problems in the continuous learning setting. By backpropagating through the changepoint estimation framework, MOCA learns both a rapidly adaptive underlying predictive model (in the form of the meta-learning model), as well as an effective changepoint detection algorithm. MOCA is a generic framework which can be paired with many existing meta-learning algorithms. We demonstrate the performance of MOCA on both regression and classification settings with unobserved task switches.

## 2    PROBLEM STATEMENT

Our goal is to apply meta-learning tools to the problem of task-unsegmented continual learning, in which an agent is presented sequentially with input $\boldsymbol{x}_t$, asked to make a (probabilistic) prediction $p(\hat{\boldsymbol{y}}_t \mid \boldsymbol{x}_t)$, and is then given the true label $\boldsymbol{y}_t$, and can thus ideally improve its predictions by learning from the labeled examples. Following the terminology of meta-learning, we assume that these data are drawn from a distribution according to some latent *task* $\mathcal{T}_t$, $p(\boldsymbol{x}_t, \boldsymbol{y}_t \mid \mathcal{T}_t) = p(\boldsymbol{x}_t \mid \mathcal{T}_t)p(\boldsymbol{y}_t \mid \boldsymbol{x}_t, \mathcal{T}_t)$. We will write $\boldsymbol{x}, \boldsymbol{y} \sim \mathcal{T}_t$ as shorthand for $\boldsymbol{x}, \boldsymbol{y} \sim p(\boldsymbol{x}, \boldsymbol{y} \mid \mathcal{T}_t)$. We assume a distribution over tasks, which we write $p(\mathcal{T})$, and that the initial task $\mathcal{T}_1 \sim p(\mathcal{T})$. At each timestep, the task is re-sampled from $p(\mathcal{T})$ with some probability $\lambda$ (which we refer to as the hazard rate), or remains the same.

Our goal is to optimize a learning agent to perform well in this setting. Let $p_{\boldsymbol{\theta}}(\hat{\boldsymbol{y}}_t \mid \boldsymbol{x}_{1:t}, \boldsymbol{y}_{1:t-1})$ by the agent's prediction for $\boldsymbol{y}_t$ given input $\boldsymbol{x}_t$ and the past labeled examples. We will evaluate the learner's performance through a negative log likelihood loss, and our objective is as follows:

$$\min_{\boldsymbol{\theta}} \quad \mathbb{E}\left[\sum_{t=1}^{\infty} -\log p_{\boldsymbol{\theta}}(\boldsymbol{y}_t \mid \boldsymbol{x}_{1:t}, \boldsymbol{y}_{1:t-1})\right]$$

$$\text{subject to} \quad \boldsymbol{x}_t, \boldsymbol{y}_t \sim \mathcal{T}_t, \quad \mathcal{T}_t = \begin{cases} \mathcal{T}_{t-1} & \text{w.p. } 1 - \lambda \\ \mathcal{T}_{t,\text{new}} & \text{w.p. } \lambda \end{cases} \tag{1}$$

$$\mathcal{T}_1 \sim p(\mathcal{T}), \quad \mathcal{T}_{t,\text{new}} \sim p(\mathcal{T})$$

We assume that we have access to a representative time series generated in the same manner from the same distribution of tasks, and use this time series to optimize $\boldsymbol{\theta}$ in an offline, meta-training phase. Critically, however, in stark contrast to standard meta-learning approaches, we do not assume access to task segmentation, i.e. that this offline data is pre-grouped by latent parameter $\mathcal{T}$. Moreover, we highlight that we consider the case of individual data points provided sequentially, in contrast to the common "$k$-shot, $n$-way" problem setting prevalent in few-shot learning (especially classification). Our setting may easily be extended to the setting in which multiple data points are observed simultaneously.

## 3    PRELIMINARIES

### 3.1    META-LEARNING

We begin by presenting a unified perspective on meta-learning in the task-segmented setting, which allows straightforward presentation of the algorithms used in this work as well as generalization to the task-unsegmented case. The core idea of meta-learning is to directly optimize the few-shot learning performance of a machine learning model over a *distribution* of learning tasks, rather than just a single task, with the goal of this learning performance generalizing to other tasks from this distribution.

A meta-learning method consists of two phases: meta-training and online adaptation. Let $\boldsymbol{\theta}$ be the parameters of this model learned via meta-training. During online adaptation, the model uses context data $\mathcal{D}_t = (\boldsymbol{x}_{1:t}, \boldsymbol{y}_{1:t})$ from within one task to compute statistics

$$\boldsymbol{\eta}_t = f_{\boldsymbol{\theta}}(\mathcal{D}_t) \tag{2}$$

where $f$ is a function parameterized by $\boldsymbol{\theta}$. For example, in MAML (Finn et al., 2017), the statistics are the neural network weights after gradient updates computed using $\mathcal{D}_t$. In neural processes (Garnelo et al., 2018), the statistics are the aggregated context parameters computed via encoding and aggregating the context data. For recurrent network-based meta-learning algorithms, these statistics correspond to the hidden state of the network. For a simple nearest-neighbors model, $\boldsymbol{\eta}$ may simply be the context data. The model then performs predictions by using these statistics to define a conditional distribution on $\boldsymbol{y}$ given new inputs $\boldsymbol{x}$,

$$\boldsymbol{y} \mid \boldsymbol{x}, \mathcal{D}_t \sim p_{\boldsymbol{\theta}}(\boldsymbol{y} \mid \boldsymbol{x}, \boldsymbol{\eta}_t).$$

Adopting a Bayesian perspective, we refer to $p_{\boldsymbol{\theta}}(\boldsymbol{y} \mid \boldsymbol{x}, \boldsymbol{\eta}_t)$ as the posterior predictive distribution.

The performance of this model on this task can be evaluated by considering how well this posterior predictive distribution matches the true task data distribution,

$$\mathcal{L}(\mathcal{D}_t, \boldsymbol{\theta}) = D(p(\boldsymbol{y} \mid \boldsymbol{x}, \mathcal{T}_i) || p_{\boldsymbol{\theta}}(\boldsymbol{y} \mid \boldsymbol{x}, f_{\boldsymbol{\theta}}(\mathcal{D}_t)))$$

where $D$ is a measure of the dissimilarity of the two distributions, e.g. the KL divergence, for which this objective becomes standard negative log likelihood minimization.

Meta-learning optimizes the parameters $\boldsymbol{\theta}$ such that the model performs well across a distribution of tasks,

$$\min_{\boldsymbol{\theta}} \ \mathbb{E}_{\mathcal{T}_i \sim p(\mathcal{T})} \left[ \mathbb{E}_{\mathcal{D}_t \sim \mathcal{T}_i} \left[ \mathcal{L}(\mathcal{D}_t, \boldsymbol{\theta}) \right] \right].$$

Across most meta-learning algorithms, including all of those referenced above, both the update rule and the prediction function are chosen to be differentiable operations, such that the parameters can be optimized via stochastic gradient descent. Given a dataset pre-segmented into groups of data from individual tasks, standard meta-learning algorithms operate via first sampling a group for which $\mathcal{T}$ is fixed, treating part of that group as the context data $\mathcal{D}_t$ and sampling from the remainder to obtain test points $(\boldsymbol{x}, \boldsymbol{y})$ from the same task. While this strategy can be very effective and produce expressive models that are capable of few-shot learning in complex domains, it relies on task segmentation which in many settings, especially continual learning, is not easily available.

## 3.2 Bayesian Online Changepoint Detection

To enable meta-learning without task segmentation, we extend prior work in changepoint detection. Specifically, we build on Bayesian online changepoint detection (Adams & MacKay, 2007), an approach for detecting changepoints (i.e. task switches) originally presented in a streaming unconditional density estimation context, which we review here.

BOCPD operates by maintaining a belief distribution over *run lengths*, i.e. how many of the past data points $\boldsymbol{y}_t$ correspond to the current task. At time $t$, run length of $r_t = \tau$ indicates that the task has switched $\tau$ timesteps ago, i.e. $\mathcal{D}_{-\tau} = \boldsymbol{y}_{t-\tau:t}$ are all drawn from a shared task $\mathcal{T}$. A belief that $r_t = 0$ implies that there has been a task switch, and that the current datapoint $\boldsymbol{y}_t$ was drawn from a new task $\mathcal{T}' \sim p(\mathcal{T})$. We denote this belief distribution at time $t$ as $b_t(r_t) = p(r_t \mid \boldsymbol{y}_{1:t-1})$.

Given $r_t$, we know the past $r_t$ data points all correspond to the same task, and thus the density $p(\boldsymbol{y}_t \mid \boldsymbol{y}_{1:t-1}, r_t)$ corresponds to the posterior predictive density after conditioning on the past $r_t$ data points. We can reason about the overall posterior predictive by marginalizing over the run length $r_t$ according to $b_t(r_t)$,

$$p(\boldsymbol{y}_t \mid \boldsymbol{y}_{1:t-1}) = \sum_{r_t=0}^{t-1} p(\boldsymbol{y}_t \mid \boldsymbol{y}_{1:t-1}, r_t) b_t(r_t),$$

where $p(\boldsymbol{y}_t \mid \boldsymbol{y}_{1:t-1}, r_t)$ is referred to as the *underlying predictive model* (UPM). BOCPD recursively computes posterior predictive densities for each value of $r_t \in \{0, \ldots, t-1\}$, and then evaluates new datapoints $\boldsymbol{y}_{t+1}$ under these posterior predictive densities to update the belief distribution $b(r_t)$. In this work, we extend this approach of Adams & MacKay (2007) beyond Bayesian unconditional density estimation to apply to general meta-learning models operating in the conditional density estimation setting, and derive these update rules in more detail for our context.

## 4 Meta-Learning via Online Changepoint Analysis

MOCA uses Bayesian changepoint detection to enable the application of meta-learning algorithms to settings without task segmentation, both at train and test time. Specifically, we extend BOCPD to derive a recursive Bayesian filtering algorithm for run length in the conditional and joint density estimation setting, and leverage a base meta-learning algorithm with parameters $\boldsymbol{\theta}$ to provide an underlying predictive model when conditioned on a run length. In the following subsections, we first derive MOCA's Bayesian filtering updates, and then outline how the full framework can be used to both train and evaluate meta-learning models on time series without task segmentation.

### 4.1 Bayesian Run-length Filtering

As in BOCPD, MOCA maintains a belief over possible run lengths $r_t$. Throughout this paper, we use $b_t$ to refer to the updated belief before observing data at that timestep, $(\boldsymbol{x}_t, \boldsymbol{y}_t)$. Note that $b_t$ is a discrete distribution with support over $r_t \in \{0, ..., t-1\}$.

At time $t$, the agent first observes the input $\boldsymbol{x}_t$, then makes a prediction $p(\boldsymbol{y}_t \mid \boldsymbol{x}_{1:t}, \boldsymbol{y}_{1:t-1})$, and subsequently observes $\boldsymbol{y}_t$. Generally, the latent task can influence both the marginal distribution of the input, $p(\boldsymbol{x}_t \mid \boldsymbol{x}_{1:t-1}, \boldsymbol{y}_{1:t-1})$ as well as the conditional distribution $p(\boldsymbol{y}_t \mid \boldsymbol{x}_{1:t}, \boldsymbol{y}_{1:t-1})$. Thus,

the agent can update its belief over run lengths once after observing the input $\boldsymbol{x}_t$, and again after observing the label $\boldsymbol{y}_t$. We will use $b_t(r_t \mid \boldsymbol{x}_t) = p(r_t \mid \boldsymbol{x}_{1:t}, \boldsymbol{y}_{1:t-1})$ to represent the updated belief over run length after observing only $\boldsymbol{x}_t$, and $b_t(r_t \mid \boldsymbol{x}_t, \boldsymbol{y}_t) = p(r_t \mid \boldsymbol{x}_{1:t}, \boldsymbol{y}_{1:t})$ to represent the fully updated belief over $r_t$ after observing $\boldsymbol{y}_t$. Finally, we will propagate this forward in time according to our assumptions on task dynamics to compute $b_{t+1}(r_{t+1})$, which is used in the subsequent timestep.

To derive the Bayesian update rules, we start by noting that the updated posterior is proportional to the joint density,

$$b_t(r_t \mid \boldsymbol{x}_t) = p(r_t \mid \boldsymbol{x}_{1:t}, \boldsymbol{y}_{1:t-1}) = Z^{-1} p(r_t, \boldsymbol{x}_t \mid \boldsymbol{x}_{1:t-1}, \boldsymbol{y}_{1:t-1})$$
$$= Z^{-1} p(\boldsymbol{x}_t \mid \boldsymbol{x}_{1:t-1}, \boldsymbol{y}_{1:t-1}, r_t) p(r_t \mid \boldsymbol{x}_{1:t-1}, \boldsymbol{y}_{1:t-1})$$
$$= Z^{-1} p_{\boldsymbol{\theta}}(\boldsymbol{x}_t \mid \boldsymbol{\eta}_{t-1}[r_t]) b_t(r_t) \tag{3}$$

where the normalization constant $Z$ can be computed by summing over the finite support of $b_{t-1}(r_t)$. Importantly, this update requires $p_{\boldsymbol{\theta}}(\boldsymbol{x}_t \mid \boldsymbol{\eta}_{t-1}[r_t])$, the base meta-learning algorithm's posterior predictive density over the inputs. Within classification, this density is available for generative models, and thus a generative approach is favorable to a discriminative approach within MOCA. In regression, it is uncommon to estimate the distribution of the independent variable. We take the same approach in this work and assume that $\boldsymbol{x}_t$ is independent of the task for regression problems, in which case $b_t(r_t \mid \boldsymbol{x}_t) = b_t(r_t)$.

Next, upon observing $\boldsymbol{y}_t$, we can similarly factor the belief over run lengths for the next timestep,

$$b_t(r_t \mid \boldsymbol{x}_t, \boldsymbol{y}_t) = Z^{-1} p_{\boldsymbol{\theta}}(\boldsymbol{y}_t \mid \boldsymbol{x}_t, \boldsymbol{\eta}_{t-1}[r_t]) b_t(r_t \mid \boldsymbol{x}_t). \tag{4}$$

Again, the normalization constant can be computed via a sum over the support of $r_t$.

Finally, we must propagate this belief forward in time to obtain $b_{t+1}(r_{t+1})$:

$$b_{t+1}(r_{t+1}) = p(r_{t+1} \mid \boldsymbol{x}_{1:t}, \boldsymbol{y}_{1:t}) = \sum_{r_t} p(r_{t+1}, r_t \mid \boldsymbol{x}_{1:t}, \boldsymbol{y}_{1:t})$$

$$= \sum_{r_t} p(r_{t+1} \mid r_t, \boldsymbol{x}_{1:t}, \boldsymbol{y}_{1:t}) p(r_t \mid \boldsymbol{x}_{1:t}, \boldsymbol{y}_{1:t}) = \sum_{r_t} p(r_{t+1} \mid r_t) b_t(r_t \mid \boldsymbol{x}_t, \boldsymbol{y}_t).$$

where we have exploited the assumption that the changes in task, and hence the evolution of run length $r_t$, happen independently of the data generation process. The conditional run-length distribution $p(r_{t+1} \mid r_t)$ is defined by our model of task evolution.

Recall that we assume that the task switches with fixed probability $\lambda$, the hazard rate. Thus, for all $r_t$, $p(r_{t+1} = 0 \mid r_t) = \lambda$, implying

$$b_{t+1}(r_{t+1} = 0) = \sum_{r_t} \lambda b_t(r_t \mid \boldsymbol{x}_t, \boldsymbol{y}_t) = \lambda. \tag{5}$$

Conditioned on the task remaining the same, $r_{t+1} = k > 0$ and $r_t = k - 1$. Thus, $p(r_{t+1} = k \mid r_t) = (1 - \lambda) \mathbb{1}\{r_t = k - 1\}$ implying

$$b_{t+1}(r_{t+1} = k) = (1 - \lambda) b_t(r_t = k - 1 \mid \boldsymbol{x}_t, \boldsymbol{y}_t). \tag{6}$$

Equations (5) and (6) together define $b_{t+1}$ over its support $r_{t+1} \in \{0, \dots, t\}$

## 4.2 META LEARNING WITHOUT TASK SEGMENTATION

By taking a Bayesian filtering approach to changepoint detection, we avoid hard assignments of changepoints and instead perform a soft selection over run lengths. In this way, MOCA is able to backpropagate through the changepoint detection and directly optimize the underlying predictive model, which may be any meta-learning model that admits a probabilistic interpretation.

MOCA processes a time series sequentially. We initialize $b_1(r_1 = 0) = 1$, and initialize the posterior statistics for $\boldsymbol{\eta}_0[r_1 = 0]$ as specified by the parameters $\boldsymbol{\theta}$ of the meta learning algorithm. Then, at timestep $t$, we first observe inputs $\boldsymbol{x}_t$ and update our belief over run length accordingly, computing $b_t(r_t \mid \boldsymbol{x}_t)$ according to (3). Next, we marginalize over this belief to make a probabilistic prediction for the label $\boldsymbol{y}_t$,

$$p_{\boldsymbol{\theta}}(\hat{\boldsymbol{y}}_t \mid \boldsymbol{x}_{1:t}, \boldsymbol{y}_{1:t-1}) = \sum_{r_t=0}^{t-1} b_t(r_t \mid \boldsymbol{x}_t) p_{\boldsymbol{\theta}}(\hat{\boldsymbol{y}}_t \mid \boldsymbol{x}_t, \boldsymbol{\eta}_{t-1}[r_t]) \tag{7}$$

We then observe the true label $\boldsymbol{y}_t$ and incur the corresponding negative log likelihood loss. We can then use this observation to update both the belief over run length, computing $b_t(r_t \mid \boldsymbol{x}_t, \boldsymbol{y}_t)$

---

**Algorithm 1** Meta-Learning via Online Changepoint Analysis: Training

---

**Require:** Training data $\boldsymbol{x}_{1:n}, \boldsymbol{y}_{1:n}$, number of training iterations $N$, initial model parameters $\boldsymbol{\theta}$
 1: **for** $i = 1$ to $N$ **do**
 2:     Sample training batch $\boldsymbol{x}_{1:T}, \boldsymbol{y}_{1:T}$ from the full timeseries.
 3:     Initialize belief over run length $b_1(r_1 = 0) = 1$
 4:     Initialize posterior statistics $\boldsymbol{\eta}_0[r = 0]$ according to $\boldsymbol{\theta}$
 5:     **for** $t = 1$ to $T$ **do**
 6:         Observe $\boldsymbol{x}_t$
 7:         Compute $b_t(r_t \mid \boldsymbol{x}_t)$ according to (3)
 8:         Predict $p_{\boldsymbol{\theta}}(\hat{\boldsymbol{y}}_t \mid \boldsymbol{x}_{1:t}, \boldsymbol{y}_{1:t-1})$ according to (7)
 9:         Observe $\boldsymbol{y}_t$
10:         Incur NLL loss $\ell_t = -\log p_{\boldsymbol{\theta}}(\boldsymbol{y}_t \mid \boldsymbol{x}_{1:t}, \boldsymbol{y}_{1:t-1})$
11:         Compute updated posteriors $\boldsymbol{\eta}_t[r_t]$ for all $r_t$ according to (8)
12:         Compute $b_t(r_t \mid \boldsymbol{x}_t, \boldsymbol{y}_t)$ according to (4)
13:         Compute updated belief over run length $b_{t+1}$ according to (6) and (5)
14:     **end for**
15:     Compute $\nabla_{\boldsymbol{\theta}} \sum_{t=k}^{k+T} \ell_t$ and perform gradient descent update to $\boldsymbol{\theta}$
16: **end for**

---

according to (4), as well as update the posterior statistics for all the run lengths using the labeled example. A recursive update rule for $\boldsymbol{\eta}$ allows these parameters to be computed efficiently using the past values of $\boldsymbol{\eta}$

$$\boldsymbol{\eta}_t[r] = h(\boldsymbol{x}_t, \boldsymbol{y}_t, \boldsymbol{\eta}_{t-1}[r-1]) \quad \forall \; r = 1, \dots, t. \tag{8}$$

While MOCA could be used with an algorithm which didn't admit such a recursive update rule, this would require storing data online and running the non-recursive posterior computation (2) on $\mathcal{D}_{-r_t}$ for every $r_t$, which involves $t$ operations using datasets of sizes from 0 to $t$, and thus can be an $O(t^2)$ operation. In contrast, the recursive updates involve $t$ operations involving just the latest datapoint, yielding $O(t)$ complexity. Finally, we propagate the belief over run length forward in time according to (5) and (6) to obtain $b_t(r_{t+1})$ to be ready to process the next data point.

Since all these operations are differentiable, given a training time series in which there are task switches $\boldsymbol{x}_{1:n}, \boldsymbol{y}_{1:n}$, we can run this procedure, sum the NLL losses incurred at each step, and use backpropagation within a standard deep learning framework to optimize the parameters of the base learning algorithm $\boldsymbol{\theta}$. Algorithm 1 outlines this training procedure. In practice, we sample shorter time-series of length $T$ from the training data to ease computational requirements during training; we discuss implications of this in the appendix. If available, a user can input various levels of knowledge on task segmentation by manually updating $b(r_t)$ at any time; further details on this task semi-segmented use case are provided in the appendix.

## 5 MAKING YOUR MOCA: MODEL INSTANTIATIONS

Thus far, we have presented MOCA at an abstract level, highlighting the fact that it can be used with any meta-learning model that admits the probabilistic interpretation as an underlying predictive model. However, there are several practical considerations in the choice of meta-learning algorithm which can influence the computational efficiency and overall performance of MOCA. For the experiments in this paper, we leverage two meta-learning algorithms which offer a clean Bayesian learning interpretation, relatively low-dimensional posterior statistics, recursive updates for these statistics, and computationally efficient likelihood evaluation under the posterior predictive. For regression experiments, we use ALPaCA (Harrison et al., 2018); for classification experiments, we use a novel algorithm based on similar Bayesian updates which we refer to as PCOC, for probabilistic clustering for online classification. For completeness, we offer a high level overview of these algorithms and show how they fit into the MOCA framework in the following subsections.

### 5.1 ALPACA: BAYESIAN META-LEARNING FOR REGRESSION

ALPaCA (Harrison et al., 2018) is a meta-learning approach for which the base learning model is Bayesian linear regression in a learned feature space $\boldsymbol{y} \mid \boldsymbol{x} \sim \mathcal{N}(K^T \boldsymbol{\phi}(\boldsymbol{x}, \boldsymbol{w}), \Sigma_\epsilon)$ where $\boldsymbol{\phi}(\boldsymbol{x}, \boldsymbol{w})$ is a feed-forward neural network with weights $\boldsymbol{w}$ mapping inputs $\boldsymbol{x}$ to a $n_\phi$-dimensional feature space. ALPaCA maintains a matrix-normal distribution over $K$, and thus, assuming Gaussian likelihood,

results in a matrix-normal posterior distribution over $K$. This posterior inference may be performed exactly, and computed recursively. The matrix-normal distribution on the last layer results in a Gaussian posterior predictive density.

We fix the prior $K \sim \mathcal{MN}(\bar{K}_0, \Sigma_\epsilon, \Lambda_0^{-1})$. In this matrix-normal prior, $\bar{K}_0 \in \mathbb{R}^{n_\phi \times n_y}$ is the prior mean and $\Lambda_0$ is a $n_\phi \times n_\phi$ precision matrix (inverse of the covariance). Given this prior and data model, the posterior may be recursively computed as follows. First, we define $Q_t = \Lambda_t^{-1}\bar{K}_t$. Then, the one step posterior update is

$$\Lambda_{t+1}^{-1} = \Lambda_t^{-1} - \frac{(\Lambda_t^{-1}\boldsymbol{\phi}(\boldsymbol{x}_{t+1}))(\Lambda_t^{-1}\boldsymbol{\phi}(\boldsymbol{x}_{t+1}))^T}{1 + \boldsymbol{\phi}^T(\boldsymbol{x}_{t+1})\Lambda_t^{-1}\boldsymbol{\phi}(\boldsymbol{x}_{t+1})} \qquad Q_{t+1} = \boldsymbol{y}_{t+1}\boldsymbol{\phi}^T(\boldsymbol{x}_{t+1}) + Q_t \qquad (9)$$

and the posterior predictive distribution is

$$p_{\boldsymbol{\theta}}(\hat{\boldsymbol{y}}_{t+1} \mid \boldsymbol{x}_{1:t+1}, \boldsymbol{y}_{1:t}) = \mathcal{N}((\Lambda_t^{-1}Q_t)^T\boldsymbol{\phi}(\boldsymbol{x}_{t+1}), (1 + \boldsymbol{\phi}^T(\boldsymbol{x}_{t+1})\Lambda_t^{-1}\boldsymbol{\phi}(\boldsymbol{x}_{t+1}))\Sigma_\epsilon). \qquad (10)$$

In summary, ALPaCA is a meta learning model for which the posterior statistics are $\boldsymbol{\eta}_t = \{Q_t, \Lambda_t^{-1}\}$, and the recursive update rule $h(\boldsymbol{x}, \boldsymbol{y}, \boldsymbol{\eta})$ is given by (9). The parameters that are meta-learned are the prior statistics, the feature network weights, and the noise covariance: $\boldsymbol{\theta} = \{\bar{K}_0, \Lambda_0, \boldsymbol{w}, \Sigma_\epsilon\}$. Note that, as is typical in regression, ALPaCA only models the conditional density $p(\boldsymbol{y} \mid \boldsymbol{x})$, implicitly assuming that $p(\boldsymbol{x})$ is independent of the underlying task.

## 5.2 PCOC: BAYESIAN META-LEARNING FOR CLASSIFICATION

In the classification setting, one can obtain a similar Bayesian meta-learning algorithm by performing Gaussian discriminant analysis in a learned feature space. This is a novel approach to meta-learning for classification which we term probabilistic clustering for online classification (PCOC, pronounced "peacock"). We present a concise description of this algorithm here but defer to the appendix for a more detailed discussion.

In PCOC we process labeled input/class pairs $(\boldsymbol{x}_t, y_t)$ by encoding the input through an embedding network $\boldsymbol{z}_t = \boldsymbol{\phi}(\boldsymbol{x}_t; \boldsymbol{w})$, and performing Bayesian density estimation for every class. Specifically, we assume a Categorical-Gaussian generative model in this embedding space, and impose the conjugate Dirichlet prior over the class probabilities and a Gaussian prior over the mean for each class,

$$y_t \sim \text{Cat}(p_1, \ldots, p_{n_y}), \qquad\qquad p_1, \ldots, p_{n_y} \sim \text{Dir}(\boldsymbol{\alpha}_0),$$

$$\boldsymbol{z}_t \mid y_t \sim \mathcal{N}(\bar{z}_{y_t}, \Sigma_{\epsilon,y_t}), \qquad\qquad \bar{z}_{y_t} \sim \mathcal{N}(\mu_{y_t,0}, \Lambda_{y_t,0}^{-1}).$$

Given labeled context data $(\boldsymbol{x}_t, y_t)$, the algorithm updates its belief over the Gaussian mean for the corresponding class, as well as its belief over the probability of each class. As with ALPaCA, these posterior computations can be performed through closed form recursive updates. Defining $\boldsymbol{q}_{i,t} = \Lambda_{i,t}\boldsymbol{\mu}_{i,t}$, we have

$$\boldsymbol{\alpha}_t = \boldsymbol{\alpha}_{t-1} + \boldsymbol{1}_{y_t} \qquad \boldsymbol{q}_{y_t,t} = \boldsymbol{q}_{y_t,t-1} + \Sigma_{\epsilon,y_t}\boldsymbol{\phi}(\boldsymbol{x}_t) \qquad \Lambda_{y_t,t} = \Lambda_{y_t,t-1} + \Sigma_{\epsilon,y_t}$$

where $\boldsymbol{1}_i$ denotes a one-hot vector with a one at index $i$. Terms not related to class $y_t$ are left unchanged in this recursive update. Given this set of posterior parameters $\boldsymbol{\eta}_t = \{\boldsymbol{\alpha}_t, \boldsymbol{q}_{1:J,t}, \Lambda_{1:J,t}\}$, the posterior predictive density in the embedding space can be computed as

$$p(y) = \alpha_{y,t}/(\sum_{i=1}^{J}\alpha_{i,t}) \qquad\qquad p(\boldsymbol{z}, y) = p(y)\mathcal{N}(\boldsymbol{z}; \Lambda_{y,t}^{-1}\boldsymbol{q}_{y,t}, \Lambda_{y,t}^{-1} + \Sigma_{\epsilon,y})$$

where $\mathcal{N}(\boldsymbol{z}; \mu, \Sigma)$ denotes the Gaussian pdf with mean $\mu$ and covariance $\Sigma$ evaluated at $\boldsymbol{z}$. Applying Bayes rule, the posterior predictive on $y_{t+1}$ given $\boldsymbol{x}_{t+1}$ is

$$p(y_{t+1} = j \mid \boldsymbol{x}_{1:t+1}, \boldsymbol{y}_{1:t}) = \frac{p(\boldsymbol{z} = \boldsymbol{\phi}(\boldsymbol{x}_t), y = j)}{\sum_{i=1}^{J} p(\boldsymbol{z} = \boldsymbol{\phi}(\boldsymbol{x}_t), y = i)}.$$

This generative modeling approach also allows computing $p(\boldsymbol{z}_{t+1} \mid \boldsymbol{\eta}_t)$ by simply marginalizing out $y$ from the joint density of $p(\boldsymbol{z}, y)$,

$$p(\boldsymbol{z}_{t+1} \mid \boldsymbol{\eta}_t) = \sum_{y=1}^{J} p(y)\mathcal{N}(\boldsymbol{z}_{t+1}; \mu_t, \Lambda_{y,t}^{-1} + \Sigma_{\epsilon,y})$$

As this only depends on the input $\boldsymbol{x}$, we can use this likelihood within MOCA to update the run length belief upon seeing $\boldsymbol{x}_t$ and before predicting $\hat{y}_t$.

In summary, PCOC performs Bayesian Gaussian discriminant analysis for online classification, and meta-learns the parameters $\boldsymbol{\theta} = \{\boldsymbol{\alpha}_0, \boldsymbol{q}_{1:J,0}, \Lambda_{1:J,0}, \boldsymbol{w}, \Sigma_{\epsilon,1:J}\}$ for efficient few-shot online classification. In practice, we assume that all the covariances are diagonal to limit memory footprint of the posterior parameters. PCOC can be thought of a Bayesian analogue of prototypical networks (Snell et al., 2017). Further details regarding PCOC can be found in the appendix.

## 6 RELATED WORK

**Online Learning, Continuous Learning, and Concept Drift Adaptation.** A substantial literature exists on online, continual and lifelong learning (Hazan, 2016; Chen & Liu, 2016). While these terms are often used interchangeably and inconsistently, they all roughly correspond to the problem of learning within a streaming series of tasks, wherein it is desirable to re-use information from previous tasks while avoiding negative transfer French (1999); Thrun & Pratt (2012). Typically, continual learning assumes access to task segmentation information, whereas online learning does not (Aljundi et al., 2019). Regularization approaches (Kirkpatrick et al., 2017; Hazan, 2016; Li & Hoiem, 2017) have been shown to be an effective method for avoiding forgetting in continual learning. By augmenting the loss function for a new task with a penalty for deviation from the parameters learned for previous tasks, the regularizing effects of a prior are mimicked; in contrast we explicitly learn a prior over task weights that is meta-trained to be rapidly adaptive. Thus, MOCA is capable of avoiding substantial negative transfer by detecting task change, and rapidly adapting to new tasks. Aljundi et al. (2019) loosen the assumption of task segmentation in continual learning and operate in a similar setting to that addressed herein, but their work still focuses on learning a single set of parameters that perform well on all tasks; in contrast, we operate in the meta-learning setting, aiming to learn parameters that accelerate online adaptation within a task.

**Meta-Learning for Continuous and Online Learning.** While continual learning techniques have mitigated forgetting in changing problem settings, large learning models have been slow to adapt to new tasks, due in part to the propensity of neural network models to overfit to small amounts of data. In response to this, there has been substantial interest in applying ideas from meta-learning to continual learning to enable rapid adaptation to new tasks. Indeed, some modern meta-learning models such as MAML (Finn et al., 2017) may be interpreted as regularization methods (Grant et al., 2018), wherein the regularization term is explicitly learned for fast adaptation. In the streaming data setting, several works (Nagabandi et al., 2019a; He et al., 2019) use a sliding window approach, wherein a small amount of recent data is used for conditioning. By not explicitly detecting task change and choosing the window length in response, these models risk suffering from negative transfer. Indeed, MOCA may be interpreted as an adaptive sliding window model, that actively infers the optimal window length. Nagabandi et al. (2019b) and Jerfel et al. (2019) aim to detect task changes via combining mean estimation of the dependent variable with MAML models. However, these models are both less expressive than MOCA (which maintains a full Bayesian posterior) and are not capable of task-unsegmented training. Instead, these models require pre-training with a meta-dataset that is segmented by task, limiting their applicability relative to MOCA.

**Empirical Bayes for Changepoint Models.** The Bayesian online changepoint framework of Adams & MacKay (2007) (which we leverage in this paper) and the similar, simultaneous work of Fearnhead & Liu (2007) have generated a substantial body of follow-on work since their publication. Due to the simplicity of these algorithms—in particular, the ability to compute closed-form posteriors as opposed to being forced to turn to approximate methods such as MCMC—many practical modifications and extensions have been developed. Of particular relevance are two works that investigate empirical Bayes for the underlying predictive model, which is a similar problem to that addressed herein. In particular, Paquet (2007) develop a forward-backward algorithm that allows closed-form max likelihood estimation of the prior for simple distributions via EM. Turner et al. (2009) derive general-purpose gradients for hyperparameter optimization within the BOCPD model. This approach is similar to our work, although we use neural network meta-learning models and rely on automatic differentiation for gradient computation.

## 7 EXPERIMENTAL RESULTS

We investigate the performance of MOCA in three problem settings: one in regression and two in classification. Our primary goal is to characterize the impact on performance of using MOCA to move from the standard *task-segmented* meta-learning setting to the task-unsegmented case. To this end, we investigate the performance of MOCA versus an "oracle" model that uses the same base meta-learning algorithm, but has access to exact task segmentation at train and test time. We additionally compare against baseline sliding window models of various window lengths, which again use the same meta-learning algorithm, but always condition on the last $n$ data points. These baselines are a competitive approach to learning in time-varying data streams (Gama et al., 2014) and have been used effectively for meta-learning in time-varying settings (see e.g. Nagabandi et al. (2019a)). Finally, we compare to a "train on everything" model, which only learns a prior and does

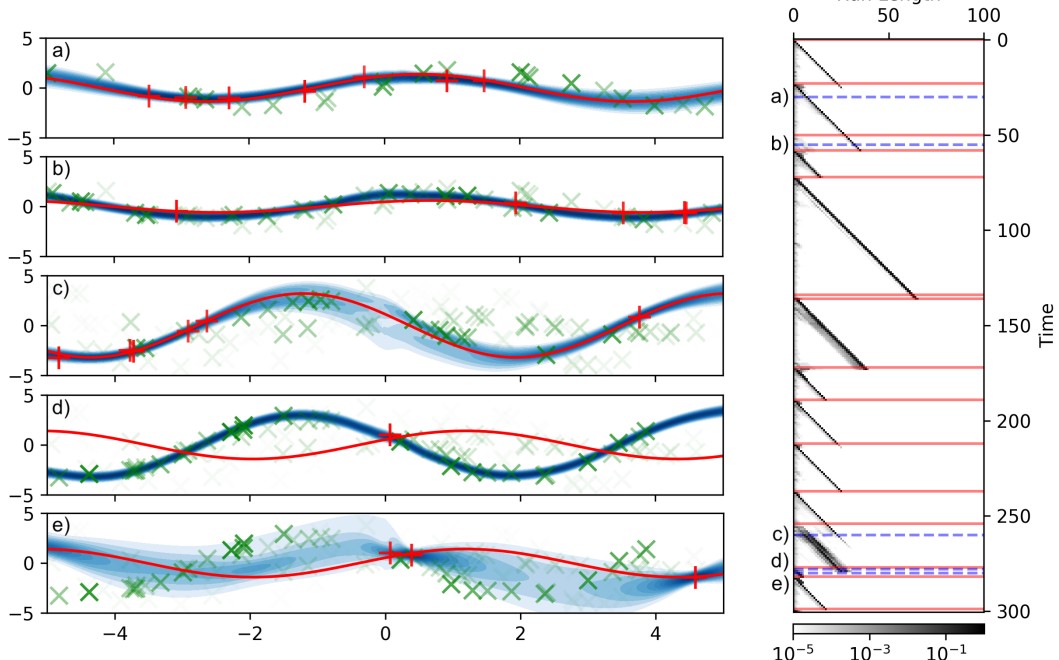

Figure 1: The performance of MOCA on the sinusoid regression problem. **Right:** The belief over run length versus time. The intensity of each point in the plot corresponds to the belief in run length at the associated time. The red lines show the true changepoints. **Left:** Visualizations of the posterior predictive density corresponding to the blue dotted lines in the figure on the right. The red line denotes the current function (task), and red points denote samples from that function. Green points denote data from previous tasks, where more faint points are older. **a)** A visualization of the posterior at an arbitrary time. **b)** The visualization of the posterior for a case in which MOCA did not successfully detect the changepoint. In this case, it is because the pre- and post-change function (corresponding to figure a and b) are highly similar. **c)** An instance of a multimodal posterior. **d)** The changepoint is initially missed due to the data generated from the post-change function being highly likely under the previous posterior. **e)** After an unlikely data point, the model increases its uncertainty as the changepoint is detected.

not adapt online, corresponding to a standard supervised learning approach. Many problems that are currently addressed with standard supervised learning in fact have underlying temporal structure that is ignored, and thus this baseline model is a valuable point of comparison.

In addition, we investigate in isolation the effects of task-segmentation information when provided at train-time and at test-time. To characterize the impact of test time segmentation, we train an oracle model and at test time, remove task segmentation and replace it with MOCA's run length estimation. We then compare this to the oracle model tested with segmentation, so the only difference is availability of test-time segmentation. Similarly, to characterize the impact of train time segmentation, we provide a model trained using MOCA with task segmentation at test time and compare this to a the same MOCA model when tested without segmentation. Finally, we investigate the performance of MOCA under partial task segmentation. Due to space constraints, we defer this to the appendix. For sinusoid experiments, confidence intervals at 95% for three different models (trained with different random seeds). For Rainbow MNIST and miniImageNet, confidence intervals are 95% for five different models.

## 7.1 SINUSOID REGRESSION

To characterize MOCA in the regression setting, we investigate the performance on a switching sinusoid problem adapted from (Finn et al., 2017), in which a task change corresponds to a re-sampled sinusoid phase and amplitude. Qualitative results are visualized for the sinusoid in Fig. 1, as well as a visualization of the belief over run length at each time. Qualitatively, MOCA is capable of accurate and calibrated posterior inference with only a handful of data points, and is capable of identifying task change extremely rapidly. Typically, it identifies task change in one timestep, if

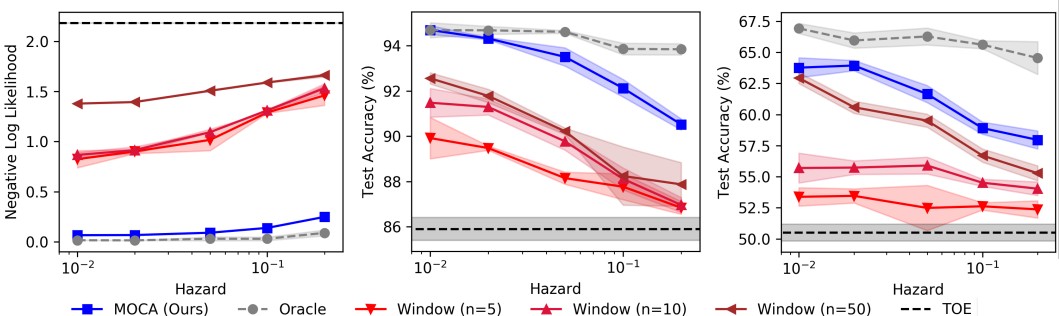

Figure 2: Performance of MOCA versus baselines in sinusoid regression (**left**; lower is better), Rainbow MNIST (**center**; higher is better), and miniImageNet (**right**; higher is better), versus hazard rate. Note that for both problems, MOCA always outperforms the baselines and the performance degrades only slightly from the performance of the oracle. In contrast, sliding window methods result in severely degraded performance.

the generated data does not happen to have high likelihood under the previous task as in Fig. 1d. Performance of MOCA versus baselines is presented in Fig. 2 for all problem domains. For sinusoid (left), MOCA achieves performance close to the oracle model and substantially outperforms the sliding window approaches for all hazard rates.

Fig. 3 shows the performance of MOCA when augmented with task segmentation at test time (violet), compared to unsegmented (blue), as well as the oracle model without test segmentation (teal) compared to with test segmentation (grey). We find that as the hazard rate increases, both the value of segmentation in training and value of segmentation at test time increases steadily. Because our regression version of MOCA is not performing density estimation for the independent variable, it is not able to detect a changepoint before incurring the loss associated with an incorrect prediction. Thus, for high changepoints, considerable loss is incurred, increasing the value of task segmentation. Interestingly and counter-intuitively, the model trained with MOCA outperforms the model trained with oracle supervision, when both are given oracle supervision at test time. The MOCA training results in a small "curriculum" effect due to the non-zero weight on placed on the prior for every training iteration; in comparison, for the oracle model, a missed prediction with a highly concentrated yet incorrect posterior occasionally results in a very large loss than may destabilize training. When the oracle model is trained with a small belief weight on the prior (even down to e.g. $10^{-16}$), the performance matches the MOCA model. This suggests that MOCA may be beneficial in training by acting as a form of curriculum.

## 7.2 RAINBOW MNIST

In the classification setting, we apply MOCA to the Rainbow MNIST dataset of Finn et al. (2019). In this dataset, MNIST digits have been perturbed via a color transformation, rotation, and scaling, and each task corresponds to a unique combination of these transformations. MOCA approaches oracle performance for most hazard rates, likely due in part to the fact that task change can usually be detected via a change in digit color. Seven colors were used, and thus with probability 6/7, MOCA has a very strong indicator of task change.

In Fig. 3, the relative effect of the MOCA train and test is visible. For high hazard rates, as expected, MOCA at test time performs slightly worse than the oracle model. The majority of performance degradation is thus due to MOCA training. Performance degradation due to MOCA training is largest for this experiment, compared to the sinusoid and miniImageNet. Because the changing digit color results in a relatively clear indicator of changepoints, and MOCA performs a belief update based on both the image and the label, MOCA performs comparably to the oracle model at test time.

## 7.3 MINIIMAGENET

Finally, we investigate the performance of MOCA on the miniImageNet benchmark task (Vinyals et al., 2016). This dataset consists of 100 ImageNet categories (Deng et al., 2009), each with 600 RGB images of resolution $84 \times 84$. In our continual learning setting, we associate each class with a semantic label that is consistent between tasks. Specifically, we split the miniImageNet dataset in to five approximately balanced high level classes, which we refer to as super-classes, as five-way classification is standard for miniImageNet (Vinyals et al., 2016; Snell et al., 2017). For example,

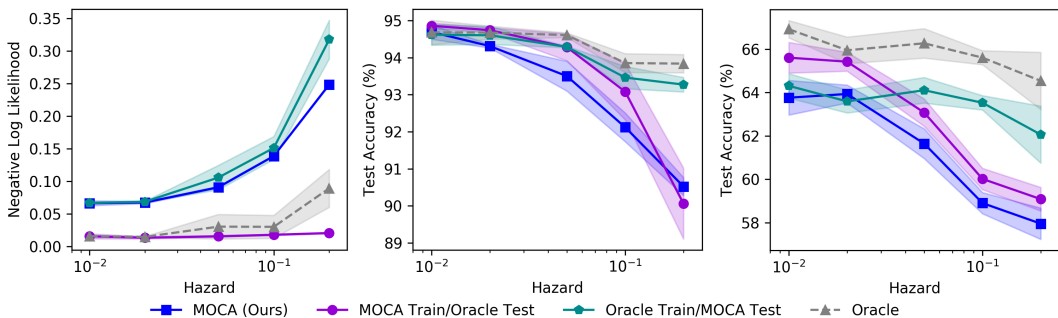

Figure 3: Performance change from augmenting a model trained with MOCA with task supervision at test time (violet) and from using changepoint estimation at test time for a model trained with task-supervision (teal), for sinusoid (**left**), Rainbow MNIST (**center**), and miniImageNet (**right**).

one super-class is dog breeds, while another is food, kitchen and clothing items; details are provided in the appendix. Then, a new task corresponds to sampling a new class within each super-class, and the problem is to classify an image as belonging to a specific super-class. This enables knowledge re-use between classes, and corresponds to a continual learning scenario in which each super-class experiences distributional shift. Note that this is somewhat different from the typical task in few-shot learning, where classes have no *a priori* semantic meaning.

Fig. 2 shows that MOCA outperforms baselines for all hazard rates. Fig. 3 shows that, in contrast to the Rainbow MNIST experiment, there is a large and constant (with respect to hazard rate) performance decrease moving from oracle to MOCA at test time. Interestingly, one would expect the performance decrease with respect to hazard rate to be attributable primarily to lack of task segmentation at test time—in fact, it appears that the trend is primarily a consequence of MOCA training. This also holds for the Rainbow MNIST experiments. This is likely a consequence of the limited amount of data, as the trend is not apparent for the sinusoid experiment.

## 8    DISCUSSION AND CONCLUSIONS

**Future Work.**    While MOCA addresses a continual learning problem setting, we have not formulated MOCA as an online learning algorithm. Specifically, MOCA meta-trains on an offline time-series, and keeps the parameters $\theta$ fixed online, whereas an online learning algorithm would not have this train/test distinction, and would consider updating $\theta$ continuously (Hazan, 2016). However, in order to do this with MOCA, we would need to keep a running buffer of all data observed so far and to use as training data to update $\theta$, which may be expensive in real-world domains where large volumes of data (e.g. high definition video from a large collection of cameras on an autonomous vehicle). Extending MOCA toward either strictly online training or a scheme to maintain an efficient replay buffer (Mnih et al., 2013; Vitter, 1985), is a promising direction of future work. Indeed, it may be possible to use MOCA's changepoint analysis to inform which data to save.

Beyond the continual learning extension, data efficiency may be improved by re-using information from previous tasks or modeling task evolution dynamics. Previous work (Nagabandi et al., 2019b; Jerfel et al., 2019; Knoblauch & Damoulas, 2018) has addressed the case in which tasks reoccur in both meta-learning and the BOCPD framework, and thus knowledge (in the form of a posterior estimate) may be re-used. In this work, we address the case in which tasks are sampled i.i.d. from a (typically continuous) distribution, and thus knowledge re-use is often impractical or adds marginal value. Broadly, moving beyond the assumption of i.i.d. tasks to task having associated dynamics (Al-Shedivat et al., 2018) represents a promising future direction.

**Conclusions.**    MOCA enables the application of existing meta-learning algorithms to problems without task segmentation, such as the problem setting of continual learning. We find that by leveraging a Bayesian perspective on meta-learning algorithms and augmenting these algorithms with a Bayesian changepoint detection scheme to automatically detect task switches within time-series, we can achieve similar predictive performance when compared to the standard task-segmented meta-learning setting, without the often prohibitive requirement of supervised task segmentation.

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

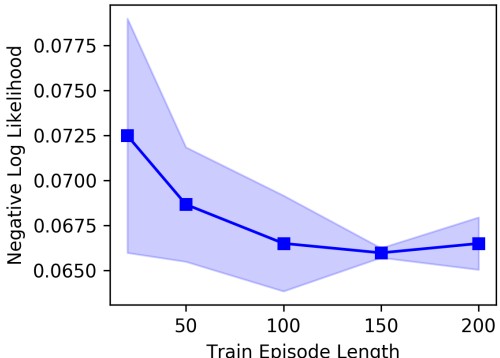

Figure 4: Performance versus the training horizon ($T$) for the sinusoid with hazard 0.01. The lowest hazard was used to increase the effects of the short training horizon. A minor decrease in performance is visible for very small training horizons (around 20), but flattens off around 100 and above. It is expected that these diminishing marginal returns will occur for all systems and hazard rates.

## A  BATCH TRAINING MOCA

In practice, we sample batches of length $T$ from the full training time series, and train on these components. While this artificially increases the observed hazard rate (as a result of the initial belief over run length being 0 with probability 1), it substantially reduces the computational burden of training. Because MOCA maintains a posterior for each possible run length, computational requirements grow linearly with $T$. Iterating over the whole training time series without any hypothesis pruning can be prohibitively expensive. While a variety of different pruning methods within BOCPD have been proposed (Wilson et al., 2010; Saatci et al., 2010), we require a pruning method which does not break model differentiability. Note that at test-time, we no longer require differentiability and so previously developed pruning methods may be applied.

Empirically, we observe diminishing marginal returns when training on longer sequences. Fig. 4 shows the performance of MOCA for varying training sequence lengths ($T$). In all experiments presented in the body of the paper, we use $T = 100$. As discussed, small $T$ values artificially inflate the observed hazard rate, so we expect to see performance improve with larger $T$ values. Fig. 4 shows that this effect results in diminishing marginal returns, with little performance improvement beyond $T = 100$. Longer training sequences lead to increased computation per iteration (as MOCA is linear in the runlength), as well as an increased memory burden (especially during training, when the computation graph must be retained by automatic differentiation frameworks). Thus, we believe it is best to train on the shortest possible sequences, and propose $T = 1/\lambda$ (where $\lambda$ is the hazard rate) as a rough rule of thumb.

## B  PROBABILISTIC CLUSTERING FOR ONLINE CLASSIFICATION

In this section we present in more detail probabilistic clustering for online classification (PCOC, pronounced "peacock"), a framework for Bayesian meta-learning for classification. PCOC extends embedding-based meta-learning algorithms (e.g. Snell et al. (2017); Vinyals et al. (2016); Allen et al. (2019); Ren et al. (2018)) to enable expressive posterior distributions (which are useful for use within the MOCA framework). However, PCOC is a valuable meta-learning algorithm outside of the MOCA framework, with many features of note for downstream applications.

PCOC maps each data point, $\boldsymbol{x}_t$, through an embedding function[1] $\boldsymbol{\phi} : \mathbb{R}^{n_x} \to \mathbb{R}^{n_\phi}$. We choose a neural network with weights $\boldsymbol{w}$ for the embedding function. We assume a generative model within the embedding space. We will assume that each class is sampled from a task dependant categorical distribution with class probabilities $p_1, \ldots, p_{n_y}$. We will assume that for each class $j$, $\boldsymbol{z} = \boldsymbol{\phi}(\boldsymbol{x})$ (for $\boldsymbol{x}_k$ with $y_k = j$) follows a Gaussian distribution with mean $\bar{\boldsymbol{z}}_j$ and variance $\Sigma_j$. This assumption is a standard generative modelling assumption corresponding to a Gaussian mixture model (Murphy, 2012). As such, our classification strategy will be based on Gaussian discriminant analysis (GDA, also referred to as quadratic discriminant analysis).

Given this Categorical-Gaussian generative model, we will fix a conjugate Dirichlet prior on class probabilities and a Gaussian prior on the class conditional mean:

$$p_1, \ldots, p_{n_y} \sim \text{Dir}(\alpha_0) \qquad\qquad \bar{z}_j \sim \mathcal{N}(\boldsymbol{\mu}_{j,0}, \Lambda_{j,0}^{-1}) \; \forall j \qquad (11)$$

---

[1]We will refer to the mapped-to space as the feature space or the embedding space.

---

**Algorithm 2** Probabilistic Clustering for Online Classification

---

**Require:** Meta-dataset $\mathcal{D}$
1: Randomly initialize weights of $\phi$, Dirichlet priors, prior mean and variance of each class
2: **while** not converged **do**
3:      **for all** $D \in \mathcal{D}$ **do**
4:          Split $D$ into conditioning data $\{\boldsymbol{x}_{1:k}, \boldsymbol{y}_{1:k}\}$ and evaluation data $\{\boldsymbol{x}_{k+1:T}, \boldsymbol{y}_{k+1:T}\}$
5:          Compute $\boldsymbol{\mu}_{j,k}, \Lambda_{j,k}$ and posterior Dirichlet concentration parameters for each class $j$
6:          Evaluate probability of evaluation data under posterior densities (Eq. 16)
7:      **end for**
8:      Update network weights and prior terms based on maximum likelihood of evaluation data
9: **end while**

---

This choice of conjugate prior and generative model means that the posterior distributions on $q$ and $\bar{z}_j$ remain Dirichlet and Gaussian respectively, and that the parameters of this posterior can be computed analytically.

The posterior parameters at time $t$ after observing $k_j$ samples of class $j$ are

$$\Lambda_{j,t} = \Lambda_{j,0} + k_j \Sigma_{\epsilon,j}^{-1}, \tag{12}$$

$$\boldsymbol{\mu}_{j,t} = \Lambda_{j,t}^{-1}(k_j \Sigma_{\epsilon,j}^{-1} \bar{\phi}_{j,t} + \Lambda_{j,0} \boldsymbol{\mu}_{j,0}), \tag{13}$$

$$\alpha_t = \sum_{j=1}^{J} k_j \mathbf{1}_j. \tag{14}$$

where $\bar{\phi}_{j,t}$ is the sample mean of the embedded points corresponding to class $j$. These can also be computed recursively, as outlined in the main paper in equation (11).

Given these posteriors, the posterior predictive distribution for class $j$ is Gaussian:

$$p(\boldsymbol{z}_{t+1} \mid y_{t+1} = j, \boldsymbol{x}_{1:t}, \boldsymbol{y}_{1:t}) = \mathcal{N}(\boldsymbol{z}_{t+1}; \boldsymbol{\mu}_{j,t}, \Lambda_{j,t}^{-1} + \Sigma_{\epsilon,j}) \tag{15}$$

Given this within-class posterior predictive, we can now consider the posterior predictive over classes. Note that, by Bayes rule,

$$p(\boldsymbol{y}_{t+1} = j \mid \boldsymbol{x}_{t+1}, \boldsymbol{x}_{1:t}, \boldsymbol{y}_{1:t}) = \frac{p(\boldsymbol{x}_{t+1} \mid \boldsymbol{y}_{t+1} = j, \boldsymbol{x}_{1:t}, \boldsymbol{y}_{1:t}) p(\boldsymbol{y}_{t+1} = j \mid \boldsymbol{x}_{1:t}, \boldsymbol{y}_{1:t})}{\sum_i p(\boldsymbol{x}_{t+1} \mid \boldsymbol{y}_{t+1} = i, \boldsymbol{x}_{1:t}, \boldsymbol{y}_{1:t}) p(\boldsymbol{y}_{t+1} = i \mid \boldsymbol{x}_{1:t}, \boldsymbol{y}_{1:t})}. \tag{16}$$

Because we have a finite number of classes, computing the partition function in the denominator is tractable. The posterior Dirichlet probabilities take the form

$$p(\boldsymbol{y}_{t+1} = j \mid \boldsymbol{x}_{1:t}, \boldsymbol{y}_{1:t}) = p(\boldsymbol{y}_{t+1} = j \mid \boldsymbol{y}_{1:t}) = \frac{\alpha_{j,t}}{\sum_i \alpha_{i,t}}, \tag{17}$$

and we choose to meta-learn the Dirichlet prior $\boldsymbol{\alpha}_0$. In the general meta-classification setting, each label has a task-dependent probability (the classes are not necessarily balanced, as is typically assumed in e.g. 1 shot and 5 shot benchmarks). As such, online estimation of the posterior allows us to infer the class probabilities within one task, and the Dirichlet priors allow us to meta-learn a belief over label probabilities between tasks. In addition to learning priors on imbalanced classes, this approach allows our model to encode confidence in the class probabilities. For example, for small $\alpha_{j,0}$'s, the model will be highly sensitive to the empirical class counts within one task, whereas for large $\alpha_{j,0}$'s, the empirical counts within one task will have a relatively small effect.

## B.1 PCOC FOR EPISODIC META-CLASSIFICATION

The standard episodic meta-classification benchmarks are typically of the form of $k$-shot (corresponding to the number of context data points observed for each class), and $n$-way (corresponding to the number of classes) (Snell et al., 2017; Finn et al., 2017; Vinyals et al., 2016). This setting is based on association between the context data and the test data; the labels do not have a priori semantic value. For example, a meta-classification problem is identical if two class labels are exchanged. This property results in simplifications for the PCOC model. Maintaining a prior over each class individually is no longer logical, as two classes with different priors could have their labels switched with no change in the problem. Therefore, in this setting, we maintain a shared prior for the mean of all classes.

This generic prior over data is useful for modified problem statements. Consider a setting in which we do not a priori know the number of classes. In this case, in which data is being provided sequentially, we wish to report if the data provided at time $t$ corresponds to a previously unobserved class. Replacing the Dirichlet prior with a Chinese restaurant process (as in e.g. Nagabandi et al. (2019b); Allen et al. (2019)) would enable a few-shot meta-classification model with a potentially expandable number of classes. Moreover, a better calibrated confidence in outputs is available, which is useful for downstream tasks.

## B.2 PCOC for Streaming Meta-Classification

We will now discuss modifications to the PCOC framework for the streaming setting. We will discuss two cases:

1. The set of labels is known a priori (and labels have semantic value—i.e. reporting class $j$ has specific meaning beyond indicating that a data point belongs to the same class as other data points of class $j$).

2. The set of labels is not known in advance, and thus our streaming meta-classification algorithm must be able to predict when a class is previously unseen.

In this paper we consider the first case. In this setting, we may directly apply PCOC as described above. Importantly, this setting allows "zero-shot" classification, which is critical in the MOCA framework, as we have distinct, semantically meaningful priors for each class mean. In the second case, the set of labels would necessarily need to be expanded over time, for which a non-parametric model may be used as described above. There are several versions of this problem statement, which is more similar to a "lifelong learning" setting, and we defer them to future work.

## B.3 Discussion

PCOC extends a line of work on meta-classification based on prototypical networks (Snell et al., 2017). This framework maps the context data to an embedding space, after which it computes the centroid for each class. For a new data point, it models the probability of belonging to each class as the softmax of the distances between the embedded point and the class centroids, for some distance metric. For Euclidean distances (which the authors focus on), this corresponds to performing frequentist estimation of class means, under the assumption that the variance matrix for each class is the identity matrix[2]. Indeed, this corresponds to the cheapest-to-evaluate simplification of PCOC. Ren et al. (2018) propose adding a class-dependent length scale (which is a scalar), which corresponds to meta-learning a frequentist estimate of the variance for each class. Moreover, it corresponds to assuming a variance that takes the form of a scaled identity matrix. Indeed, assuming diagonality of the covariance matrix results in substantial performance improvement as the matrix inverse may be performed element-wise. This reduces the numerical complexity of this operation in the (frequently high-dimensional) embedding space from cubic to linear. However, in our implementation of MOCA, we assume diagonal covariances throughtout, resulting in comparable computational complexity to the different flavors of prototypical networks. If one were to use dense covariances, the computational performance decreases substantially (due to the necessity of matrix inversions), especially in high dimensional embedding spaces.

In contrast to this previous work, PCOC has several desirable features. First, both Snell et al. (2017) and Ren et al. (2018) make the implicit assumption that the classes are balanced, whereas we perform online estimation of class probabilities via Dirichlet posterior inference. Beyond this, our approach is explicitly Bayesian, and we maintain priors over the parameters that we estimate online. This is critical for utilization in the MOCA framework. Existence of these priors allows "zero-shot" learning—it enables a model to classify incoming data to a certain class, even if no data belonging to that class has been observed within the current task. Finally, because the posteriors concentrate (the predictive variance decreases as more data is observed), we may better estimate when a change in the task has occurred. We also note that maximum likelihood estimation of Gaussian means is dominated by the James-Stein estimator (Stein, 1956), which shrinks the least squares estimator toward some prior. Moreover, the James-Stein estimator paired with empirical Bayesian estimation of the prior—which is the basis for Bayesian meta-learning approaches such as ALPaCA and PCOC—has been shown to be a very effective estimator in this problem setting (Efron & Morris, 1973).

---

[2]Snell et al. (2017) discuss this correspondence, as they outline how the choice of metric corresponds to a different assumptions on the distributions in the embedding space.

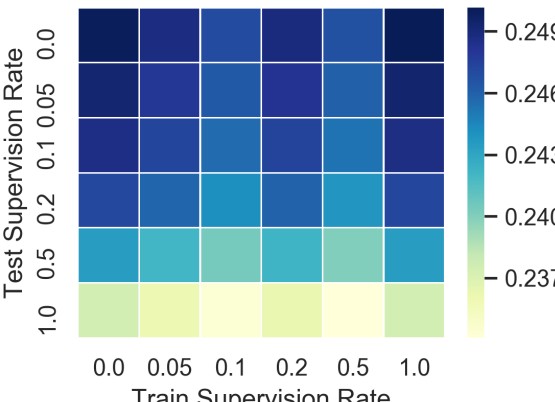

Figure 5: Test negative log likelihood of MOCA on the sinusoid problem with partial task segmentation. The partial segmentation during training results in negligible performance increase, while partial supervision at test time uniformly improves performance. Note that each column corresponds to one trained model, and thus the randomly varying performance across train supervision rates may be explained by simply results of minor differences in individual models.

## C  MOCA WITH PARTIAL TASK SEGMENTATION

Since MOCA explicitly reasons about a belief over run-lengths, it can operate anywhere in the spectrum of the task-unsegmented case as presented so far, to the fully task-segmented setting of standard meta-learning. At every time step $t$, the user can override the belief $b_t(r_t)$ to provide a degree of supervision. At known changepoints, for example, the user can override $b_t(r_t)$ to have all its mass on $r_t = 0$. If the task is known *not* to change at the given time, the user can set the hazard probability to 0 when updating the belief for the next timestep. If a user applies both of these overrides, it amounts to effectively sidestepping the Bayesian reasoning over changepoints and revealing this information to the meta-learning algorithm. If the user only applies the former, the user effectively indicates to the algorithm when known changepoints occur, but the algorithm is free to propagate this belief forward in time according to the update rules, and detect further changepoints that were not known to the user. Finally, the Bayesian framework allows a supervisor to provide their belief over a changepoint, which may not have probability mass entirely at $r_t = 0$. Thus, MOCA flexibly incorporates any type of task supervision available to a system designer.

Fig. 5 shows the performance of partial task segmentation at both train and test for the sinusoid problem, for the hazard rate 0.2. This problem was chosen as the results were highly repeatable and thus the trend is more readily observed. Here, we label a changepoint with some probability, which we refer to as the supervision rate. We do not provide supervision for any non-changepoint timesteps, and thus a supervision rate of 1 corresponds to labeling every changepoint but is not equivalent to the oracle. Specifically, the model may still have false positive changepoints, but is incapable of false negatives. This figure shows that the performance monotonically improves with increasing train supervision rate, but is largely invariant under varying train supervision. This performance improvement agrees with Fig. 3, which shows that for the sinusoid problem, performance is improved by full online segmentation. Indeed, these results show that training with MOCA results in models with comparable test performance to those with supervised changepoints, and thus there is little marginal value to task segmentation during training.

## D  COMPUTATIONAL PERFORMANCE

Fig. 6 shows the computational performance at test time on the sinusoid problem. Note that the right hand side of the curve shows a linear trend that is expected from the growing run length belief vector. However, even for 25000 iterations, the execution time is approximately 7ms for one iteration. These experiments were performed on an Nvidia Titan Xp GPU. Interestingly, on the left hand side of the curve, the time per iteration is effectively constant until the number of iterations approaches approximately 4500. Based on our code profiling, we hypothesize that this is an artifact of overhead in matrix multiplication computations done on the GPU.

Figure 6: Time per iteration versus iteration number at test time. Note that the right hand side of the curve shows the expected linear complexity expected of MOCA. Note that for these experiments, no hypothesis pruning was performed, and thus at test time performance could be constant time as opposed to linear. This figure shows 95% confidence intervals for 10 trials, but the repeatability of the computation time is consistent enough that they are not visible.

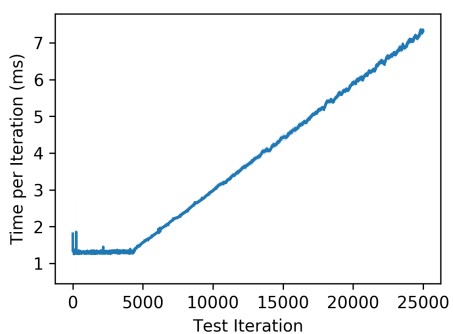

# E  EXPERIMENTAL DETAILS

## E.1  SINUSOID

To test the performance of the MOCA framework combined with ALPaCA for the regression setting, we investigate a switching sinusoid regression problem. The standard sinusoid regression problem, in which randomly sampled phase and amplitude constitute a task, is a standard benchmark in meta-learning (Finn et al., 2017). Moreover, a switching sinusoid problem is a popular benchmark in continuous learning (He et al., 2019; Javed & White, 2019). Each task consists of a randomly sampled phase in the range $[0, \pi]$ and amplitude in $[0.1, 5]$. This task was investigated for varying hazard rates. For the experiments in this paper, samples from the sinusoid had additive zero-mean Gaussian noise of variance 0.05.

## E.2  RAINBOW MNIST

The Rainbow MNIST dataset (introduced in Finn et al. (2019)) contains 56 different color/scale/rotation transformations of the MNIST dataset, where one transformation constitutes a task. We split this dataset into a train set of 49 transformations and a test set of 7. For hyperparameter optimization, we split the train set into a training set of 42 transformations and a validation of 7. However, because the dataset represents a fairly small amount of tasks (relative to the sinusoid problem, which has infinite), after hyperparameters were set we trained on all 49 tasks. We found this notably improved performance. Note that the same approach was used in Snell et al. (2017).

## E.3  MINIIMAGENET

We use the miniImageNet dataset of Vinyals et al. (2016), a standard benchmark in few-shot learning. However, the standard few-shot learning problem does not require data points to be assigned to a certain class label. Instead, given context data, the goal is to associated the test data with the correct context data. We argue that this problem setting is implausible for the continual learning setting: while observing a data stream, you are also inferring the set of possible labels. Moreover, after a task change, there is no context data to associate a new point with. Therefore we instead assume a known set of classes. We group the 100 classes of miniImageNet in to five super-classes, and perform five-way classification given these. These super-classes vary in intra-class diversity of sub-classes: for example, one of the super-class is entirely composed of sub-classes that are breeds of dogs, while another corresponds to buildings, furniture, and household objects. Thus, the strength of the prior information for each super-class varies. Moreover, the intra-class similarities are quite weak, and thus generalization from the train set to the test set is difficult and few-shot learning is still necessary and beneficial. The super-classes are detailed in table **??**.

The super-classes are roughly balanced in terms of number of classes contained. Each task correspond to sampling a class from within each super-class, which was fixed for the duration of that task. Each super-class was sampled with equal probability.

## E.4  BASELINES

Three baselines were used, described below:

| Class | Description | Train/Val/Test | Synsets |
|---|---|---|---|
| 1 | Non-dog animals | Train | n01532829, n01558993, n01704323, n01749939, n01770081, n01843383, n01910747, n02074367, n02165456, n02457408, n02606052, n04275548 |
| | | Validation | n01855672, n02138441, n02174001 |
| | | Test | n01930112, n01981276, n02129165, n02219486, n02443484 |
| 2 | Dogs, foxes, wolves | Train | n02089867, n02091831, n02101006, n02105505, n02108089, n02108551, n02108915, n02111277, n02113712, n02120079 |
| | | Validation | n02091244, n02114548 |
| | | Test | n02099601, n02110063, n02110341, n02116738 |
| 3 | Vehicles, musical instruments, nature/outdoors | Train | n02687172, n02966193, n03017168, n03838899, n03854065, n04251144, n04389033, n04509417, n04515003, n04612504, n09246464, n13054560 |
| | | Validation | n02950826, n02981792, n03417042, n03584254, n03773504, n09256479 |
| | | Test | n03272010, n04146614 |
| 4 | Food, kitchen equipment, clothing | Train | n02747177, n02795169, n02823428, n03047690, n03062245, n03207743, n03337140, n03400231, n03476684, n03527444, n03676483, n04596742, n07584110, n07697537, n07747607, n13133613 |
| | | Validation | n03770439, n03980874 |
| | | Test | n03146219, n03775546, n04522168, n07613480 |
| 5 | Building, furniture, household items | Train | n03220513, n03347037, n03888605, n03908618, n03924679, n03998194, n04067472, n04243546, n04258138, n04296562, n04435653, n04443257, n04604644, n06794110 |
| | | Validation | n02971356, n03075370, n03535780 |
| | | Test | n02871525, n03127925, n03544143, n04149813, n04418357 |

Table 1: Our super-class groupings for miniImageNet experiments.

- **Train on Everything**: This baseline consists of ignoring task variation and treating the training timeseries as one dataset. Note that many datasets contain latent temporal information that is ignored, and so this approach is effectively common practice.
- **Oracle**: In this baseline, the same ALPaCA and PCOC models were used as in MOCA, but with exact knowledge of the task switch times. Note that within a regret setting, one typically compares to the best achievable performance. The oracle actually outperforms the best achieveable performance in this problem setting, as it takes at least one data point (and the associated prediction, on which loss is incurred) to become aware of the task variation.
- **Sliding Window**: The sliding window approach is commonly used within problems that exhibit time variation, both within meta-learning (Nagabandi et al., 2019a) and continual learning (He et al., 2019; Gama et al., 2014). In this approach, the last $n$ data points are used for conditioning, under the expectation that the most recent data is the most predictive of the observations in the near future. Typically, some form of validation is used to choose the window length, $n$. As MOCA is performing a form of adaptive windowing, it should ideally outperform any fixed window length. We compare to three window lengths ($n = 5, 10, 50$), each of which are well-suited to part of the range of hazard rates that we consider.

### E.5 TRAINING DETAILS

**Sinusoid.** A standard feedforward network consisting of two hidden layers of 128 units was used with ReLU nonlinearities. These layers were followed by a 32 units layer and another tanh nonlinearity. Finally, the output layer (for which we learn a prior) was of size $32 \times 1$. The same architecture was used for all baselines. This is the same architecture for sinusoid regression as was used in Harrison et al. (2018) (with the exception of using ReLU nonlinearities instead of all tanh nonlinearities). The following parameters were used for training:

- Optimizer: Adam (Kingma & Ba, 2015)

- Learning rate: 0.02
- Batch size: 50
- Batch length: 100
- Train iterations: 7500

Batch length here corresponds to the number of timesteps in each training batch. Note that longer batch lengths are necessary to achieve good performance on low hazard rates, as short batch lengths artificially increase the hazard rate as a result of the assumption that each batch begins with a new task. The learning rate was decayed every 1000 training iterations.

We allowed the noise variance to be learned by the model. This, counter-intuitively, resulted in a substantial performance improvement over a fixed (accurate) noise variance. This is due to a curriculum effect, where the model early one increases the noise variance and learns roughly accurate features, followed by slowly decreasing the noise variance to the correct value.

**Rainbow MNIST.** In our experiments, we used the same architecture as was used as in Snell et al. (2017); Vinyals et al. (2016). It is often unclear in recent work on few-shot learning whether performance improvements are due to improvements in the meta-learning scheme or the network architecture used (although these things are not easily disentangled). As such, the architecture we use in this experiment provides fair comparison to previous few-shot learning work. This architecture consists of four blocks of 64 $3 \times 3$ convolution filters, followed by a batchnorm, ReLU nonlinearity and $2 \times 2$ max pool. On the last conv black, we removed the batchnorm and the nonlinearity. For the $28 \times 28$ Rainbow MNIST dataset, this encoder leads to a 64 dimensional embedding space. For the "train on everything" baseline, we used the same architecture followed by a fully connected layer and a softmax. This architecture is standard for image classification and has a comparable number of parameters to our model.

We used a diagonal covariance factorization within PCOC, substantially reducing the number of terms in the covariance matrix for each class and improving the performance of the model (due to the necessary inversion of the posterior predictive covariance). We learned a prior mean and variance for each class, as well as a noise covariance for each class (again, diagonal). We also fixed the Dirichlet priors to be large, effectively imbuing the model with the knowledge that the classes were balanced. The following parameters were used for training:

- Optimizer: Adam
- Learning rate: 0.02
- Batch size: 10
- Batch length: 100
- Train iterations: 5000

The learning rate was decayed every 1500 training iterations.

**miniImageNet.** Finally, for miniImageNet, we used six convolution blocks, each as previous described. This resulted in a 64 dimensional embedding space. We initially attempted to use the same four-conv backbone as for Rainbow MNIST, but the resulting 1600 dimensional embedding space had unreasonable memory requirements for batches lengths of 100. Again, for the "train on everything" baseline, we used the same architectures with one fully connected layer followed by a softmax. The following parameters were used for training:

- Optimizer: Adam
- Learning rate: 0.002
- Batch size: 10
- Batch length: 100
- Train iterations: 3000

The learning rate was decayed every 1000 training iterations. We used the validation set to monitor performance, and as in Chen et al. (2019), we used the highest validation accuracy iteration for test. We also performed data augmentation as in Chen et al. (2019) by adding random reflections and color jitter to the training data.

E.6    TEST DETAILS.

For all problems, a test horizon of 400 was used. Again, the longest possible test horizon was used to avoid artificial distortion of the test hazard rate. Both both problems, a batch of 200 evaluations was performed, and all confidence intervals correspond to 95%.

