# OpenReview forum: "Continuous Meta-Learning without Tasks"
_ICLR.cc/2020/Conference — Reject_

### Official Review · AnonReviewer3 · 2019-10-17
**Official Blind Review #3**

**Rating:** 3

**Review:**

The paper considers the meta-learning in the task un-segmented setting and apply bayesian online change point detection with meta-learning. The task un-segmented is claimed to exist in real applications and the paper explains the idea in a clear way.

My major concerns and questions are the following:

1) In Eq(4), it requires a computation of normalization constant that needs to sum over support of r_t. The support gets larger and larger with the length of sequence increases over time. Does this method scale well with long sequence?

2) The part I feel most confused are the experiments.  The paper tries to solve a problem in the task un-segmented setting, but why there is no such setting in experiments that the meta-training set cannot be segmented to different tasks?

3) The Figure 1 is hard to read. What do the red and green points in the left panel mean? The hazard rate is not defined before the experiment which makes Figure 3 also hard to understand.

4) What is the meta-learning algorithm used in the experiements, MAML, NP or RNN-based methods?  It claims MOCA can be used with any meta-learning algorithms and how does it show in experiments?

5) In Rainbow MNIST, why in high hazard rates, all models perform comparable to “train on everything”? Does it mean meta-learning does not work in this configuration and why? Does “train on everything” includes fine-tuning on the meta-test training set?

6) In Mini-IMAGENET, what does it mean by "we associate each class with a semantic label that is consistent between tasks”? Why it needs to form “super-class”?

I think the proposed method can be useful in the task un-segmented setting. But before the above-mention questions are solved and the experimental section gets more clear, I will give a conservative rating.


######################
Post-rebuttal review:
Thanks for the authors' feedback and it resolves some confusion parts in the paper. Though the author claims the reason of experiments setting, I believe it is necessary to have one experiment where task-segmentation is impossible and compare with standard sequential learning methods in that setting. Otherwise it remains a question whether the proposed method works only when the problem basically has task-segmentation.  And if the problem has task-segmentation, why not using traditional meta-learning methods, as shown as oracle methods with better performance in paper?
Since the major contribution of the paper is providing a meta-learning method to work in the problems where task-segmentation is unavailable, not having an experiment in this setting (withholding segmentation information does not exactly fall in this setting because it adds a condition that the task-segmentation information is originally accessible) is a major reason for my current evaluation.  I recognize the careful design of the MOCA and agree with some positive points raised by other reviewers. I would not be bothered if the paper is accepted while I tend to maintain the current rating because of the above-mentioned concern.


**Experience Assessment:**

I have read many papers in this area.

**Review Assessment: Checking Correctness Of Derivations And Theory:**

I assessed the sensibility of the derivations and theory.

**Review Assessment: Checking Correctness Of Experiments:**

I carefully checked the experiments.

**Review Assessment: Thoroughness In Paper Reading:**

I read the paper at least twice and used my best judgement in assessing the paper.

---

> ### Author Response · Authors · 2019-11-12
> **Response to Review 3**
>
> We thank the reviewer for the comments. First, we have discussed the growing computational complexity of the changepoint estimation in our comment to all the reviewers, above.
>
> Other responses to your comments:
>
> — Task segmentation for experiments
> We have chosen experiments that are standard benchmarks in meta-learning and few shot learning, to provide the best possible comparison with that literature. Moreover, we have deliberately chosen experiments for which task segmentation is available so that we may compare to our “oracle” model, and explicitly quantify the performance degradation when this information is withheld from the learner. Critically, we find that MOCA, without task-segmentation information, performs nearly just as well as an oracle model trained with task-segmentation, demonstrating the practicality of task-free meta-learning.
>
> — Figure 1 and Figure 3 interpretability
> Thank you for noting this. In Figure 1, the red points correspond to data points drawn from the task that the time series is currently in (they correspond to the visualized sinusoid). The green points correspond to old tasks. The model does not have access to the information of which points belong to the current task, and which belong to a previous task. Thus, in the best case, the model would ignore green points and compute the best possible posterior based on the red points. We have modified the caption of each figure to clarify this.
>
> We have clarified the discussion of the hazard throughout the paper.
>
> — Meta-learning model used
> In our regression experiments, we use ALPaCA [1], a Bayesian, optimization-based meta-learning approach that is conceptually similar to MAML. In classification, we use PCOC, a Bayesian meta-learning approach that draws similarities to prototypical networks, which is also presented in the paper. Both are described in Section 5, and a more detailed discussion of PCOC is provided in the appendix.
>
> — High hazard rate performance
> As the hazard rate increases, the number of labeled examples available within each class (for a given task) becomes smaller. For example, for a hazard rate of 0.2, in Rainbow MNIST, each class will in expectation have fewer than one example. Thus, as the hazard rate increases to 1, the data approaches being sampled iid from the union of all tasks, which exactly corresponds to the “train on everything” setting. In particular, the “train on everything” model treats the time series data as if they were iid draws from some training dataset, without any meta-learning or fine-tuning. Thus, we expect the performance of the MOCA and sliding windows models to at best be equivalent to the “train on everything” baseline as the hazard rate approaches 1.
>
> Generally, we note that we do not consider the common “k-shot, n-way” problem setting, but instead consider a constant stream of individual data points. In our setting, there is no concept of a meta-test training set; there is no conditioning data at test time. Instead, a model must use the previously observed data (with no indication of the task to which it belongs) to meta-learn.
>
> — MiniImageNet super-classes
> This is discussed in detail in appendix B. Briefly: our problem setting is a streaming classification task, in which individual images are presented sequentially, for a full time series. As such, two possible scenarios arise: either we know all possible image labels, or we have to infer the set of possible image labels online. In this work, we consider the first case. Thus, we assume the labels are known before test time. Stepping back, we can see that we have knowledge of the space of possible labels (and training examples) but do not have test-time context data. Thus, we wish to learn a prior for each super-class that can be adjusted at test time to sit a specific task.
>
> This modification to the standard few shot classification problem setting required our modified version of miniImageNet. We wish to emphasize that this is not the only possible problem statement: there are many other versions that are plausible.
>
> [1] James Harrison, Apoorva Sharma, and Marco Pavone. Meta-learning priors for efficient online Bayesian regression. Workshop on the Algorithmic Foundations of Robotics (WAFR), 2018.

---

### Official Review · AnonReviewer1 · 2019-10-23
**Official Blind Review #1**

**Rating:** 6

**Review:**

This paper pushes meta-learning towards task-unsegmented settings. Different from the traditional offline meta-learning phase with explicit task segmentation, MOCA adopts a Bayesian changepoint estimation scheme for task change detection. The setting is novel and deserves research in-depth, and the idea is easy to understand. The proposed method can learn the meta-learning model and changepoint detection model simultaneously. Besides, the MOCA framework is not designed specifically for one algorithm and can be easily combined with other meta-learning models.

However, I got some questions about this paper:
Q1: Is the ‘Hazard’ in the experiment the same to $\lambda$ in eq.1? I think notations should be consistent if they are the same.
Q2: Can other changepoint models be compared in the experiment? I found many in the related work.
Q3: The running times in Figure 2 should be reported to demonstrate the efficiency of MOCA, since the method is proposing online streams, efficiency should be promised for quickly processing.
Q4: What is ‘T’ in Algorithm 1? Are experiment results sensitive to it? Experiments about this should be conducted and reported.

Lastly, I think [1] should be cited as related work about continual learning for proposing task-free continual learning, which is very similar to the setting in this paper.
[1] Rahaf Aljundi, Klaas Kelchtermans, Tinne Tuytelaars. Task-Free Continual Learning. CVPR 2019

**Experience Assessment:**

I have published one or two papers in this area.

**Review Assessment: Checking Correctness Of Derivations And Theory:**

N/A

**Review Assessment: Checking Correctness Of Experiments:**

I assessed the sensibility of the experiments.

**Review Assessment: Thoroughness In Paper Reading:**

I read the paper at least twice and used my best judgement in assessing the paper.

---

> ### Author Response · Authors · 2019-11-12
> **Response to Review 1**
>
> Thank you for the helpful and constructive feedback. Our revised paper addresses your comments, and in particular:
>
> — Hazard rate
> Yes, the hazard rate in the experiments is that same as in equation 1. We have clarified this in Section 2.
>
> — Other changepoint models
> While other models could conceivably be used instead of the BOCPD algorithm, this approach has numerous features that we believe make it the ideal candidate for use within MOCA:
>
> First, BOCPD is an online changepoint detection algorithm, which we require at test-time as the system must reason about possible changepoints as it observes a stream of data. Following the meta-learning paradigm of training through the test-time algorithm, we use BOCPD at train time as well to avoid possible performance degradation from moving to a different changepoint detection scheme at test time.
>
> Because it is Bayesian, and maintains a belief over run length as opposed to a point estimate of changepoints, BOCPD is differentiable. This differentiability is essential for training the underlying meta-learning model, as gradients on the observed losses must be backpropagated through the changepoint detection scheme to the underlying meta-learning model. Additionally, many existing Bayesian changepoint detection algorithms (e.g. [1], [2]) do not directly provide closed-form posterior densities, and instead only offer samples from the associated posterior. BOCPD’s closed-form posterior is essential for the MOCA framework, as it provides a training objective for minimizing negative log likelihood.
>
> — Efficiency of online performance
> We have evaluated this, and the results are available in the appendix. We have found that after 25000 iterations, the time per episode is still only approximately 7 milliseconds. For true online operation (over potentially millions of data points), there are many possible approaches to hypothesis pruning that reduce computational complexity to constant time. These are discussed in our reply to all reviewers, above.
>
> — Sensitivity to T
> we have evaluated the sensitivity of the model performance to the length of the train sequence (T), available in Figure 4. We found that for T greater than approximately 100 (which is 1/hazard for the experiment), there is little to no additional training performance improvement.
>
> Similarly to training other sequence models, training on long sequences is potentially problematic for several reasons. First, long sequences require large amount of memory utilization, especially when computing gradients during training. Second, backpropagating through long sequences potentially results in exploding or vanishing gradients. For these reasons combined with our empirical evaluation, we believe training on relatively short sequences is justified.
>
> — Citation of Aljundi et al., 2019
> Thank you for this very helpful reference. We have added a discussion of this excellent work in the related work section. Their paper, as with ours, aims to remove the notion of task segmentation from continual learning. However, it does not focus on meta-learning models as the underlying model. In our work, learning occurs on two time scales: we slowly learn prior parameters, which aid in rapid fine-tuning within each task. We do not, in this work, directly address the question of continuous learning of the prior (and features). Instead, we focus on training on a task-unsegmented time series. Thus, while the data generation setting is similar, the learning objective is distinct. Indeed, a promising avenue of future work is combining these problem statements and approaches.
>
> [1] Barry, Daniel, and John A. Hartigan. "A Bayesian analysis for change point problems." Journal of the American Statistical Association. 1993.
>
> [2] Carlin, Bradley P., Alan E. Gelfand, and Adrian FM Smith. "Hierarchical Bayesian analysis of changepoint problems." Journal of the Royal Statistical Society: Series C (Applied Statistics). 1992.

---

### Official Review · AnonReviewer2 · 2019-10-23
**Official Blind Review #2**

**Rating:** 8

**Review:**

The paper proposes a solution for using meta-learning methods without the need of determining the task segmentation a priori. Authors identify that the task segmented setting is very similar to the problem of change-point detection (CPD) and particularly, they connect the generative model of the meta-learning approach to the Bayesian recursive method of Adams and MacKay 2007. The presentation of the meta-learning problem is well done and I understand its importance within the difficulties for modelling new unobserved tasks. The notation and description of the generative model included in the problem statement section is clearly understandable for any reader, this is a very positive point. They demonstrate a deep comprehension of the BOCPD model of Adams, and its extension for the meta-learning approach is original to me. Lastly, the presentation of the MOCA meta-learning algorithm is useful for reproducibility and I see it completely applicable to other scenarios. There is a noticeable effort of describing the solution for both regression and classification problems and the empirical results conclude a positive performance of MOCA.

Overall, I consider that the paper is well-written with thorough explanations. Of course, there are some details that could be improved (I will comment this later). If done, I would be willing to increase my score. The contribution of the paper is significant to meta-learning models and I think it will help to spread the Adams’ model to other type of problems.

There is an important contribution in the paper that I have to mention and it is also relevant for the future application of the Adams model. If one reads the original BOCPD paper, in particular Eq (1), where the predictive posterior p(x_t+1|x_{1:t}) is defined from a marginalisation over the run length values, it is noticeable that this equation is not used in the final recursion of the CPD method. This is because the posterior p(r_t|x_{1:t}) is sufficient for determining if there is a CP on the t time-step or not. So the predictive posterior is never used in practice (in the original paper). When I first read Adams 2007, this detail was clear to me. Surprisingly, I find that the authors have find a practical use of this equation (Eq. (7)) and it is in the main core of the meta-learning algorithm, this is fantastic.

The details I think should be improved are:

- The conditional run-length prior p(r_t|r_{t-1}) barely appears and without any detailed description is not obvious for non-familiar readers with the Bayesian CPD approach.
- The first time I read the manuscript, the use of z_t in the BOCPD presentation made me feel a bit lost. Why not use x or y? At least specify that is a toy variable for the explanation. Later on, authors change again to the x,y notation.
- Equations 3 and 4 are too similar, this looks a bit repetitive. Why not reusing one of them or say the change from one term to another?
- If reading the literature of the BOCPD and posterior extensions, the likelihood model of the detector is often referred as the underlying predictive model (UPM). Using a similar term would help for orienting familiar readers into the solution.
- As authors should have noted on their experiments, as one makes the run-length higher, the number of parameters \eta[r_t] increases. A clear state on how this is solved would help.
- In the PCOC subsection, I find the definition of y \sim q(y) a bit weird. Using Cat(_) or Multinomial likelihood notation would be a bit better.


**Experience Assessment:**

I have published one or two papers in this area.

**Review Assessment: Checking Correctness Of Derivations And Theory:**

I assessed the sensibility of the derivations and theory.

**Review Assessment: Checking Correctness Of Experiments:**

I did not assess the experiments.

**Review Assessment: Thoroughness In Paper Reading:**

I read the paper at least twice and used my best judgement in assessing the paper.

---

> ### Author Response · Authors · 2019-11-12
> **Response to Review 2**
>
> Thank you for your helpful comments. We have addressed your comments as follows:
>
> - We have clarified the discussion of the run length prior in Section 2, and how this relates to the hazard rate
> - We have moved to using y in the BOCPD background
> - We have simplified equation 4
> - We now refer to the meta-learning model as the UPM, to better match Adams and Mackay (2007)
> - We have changed the writing in the discussion of PCOC in Section 5 to be more clear (in particular, changing the q distribution to a Cat(...) distribution)
>
> All of these changes are now reflected in the paper.
>
> Finally, we have included a discussion on the computational complexity of BOCPD in our comment above.

---

### Author Response · Authors · 2019-11-12
**Response to All Reviewers**

We thank all the reviewers for their helpful responses. We have chosen to place a handful of points that are relevant to all reviewers in a general comment.

— Increased performance and better uncertainty quantification in experiments
Previously, results were reported for a single model, tested on 200 episodes of length 400. Statistics were reported over those 200 trials. We believe that this did not capture possible variation between models, and so we now train models with multiple random seeds (3 for sinusoid, 5 for Rainbow MNIST and miniImageNet) and report mean performance and 95% confidence intervals across mean performance within each model in Figure 2.

We optimized the hyperparameters and training of each experiment. As a result, performance for all models has improved substantially, especially for high hazard rates. The training details are provided in the appendix. The effect of MOCA training (versus oracle) and testing is also now more clearly visualized (Figure 3), and this is discussed in the paper.

— Computational complexity of the BOCPD changepoint estimation approach
We have empirically characterized the computational complexity of MOCA, and a plot of run time per iteration at test versus iteration is available in the appendix. We find that after 25000 iterations, the time per iteration is still only approximately 7 milliseconds. Moreover, there are a wide variety of hypothesis pruning approaches (see e.g. [1,2] in addition to the original BOCPD paper [3]).

While hypothesis pruning is directly applicable to MOCA at test time, using these methods during training without breaking differentiability remains an open problem, and we defer this to future work. We present a hyperparameter evaluation of the performance of models versus the length of the training sequence (referred to as T in the paper) and found that there were diminishing marginal returns around the T=1/(hazard rate) mark (Figure 4).

[1] Saatçi, Yunus, Ryan D. Turner, and Carl Edward Rasmussen. "Gaussian Process Change Point Models." ICML. 2010.

[2] Wilson, Robert C., Matthew R. Nassar, and Joshua I. Gold. "Bayesian online learning of the hazard rate in change-point problems." Neural computation. 2010.

[3] Adams, Ryan Prescott, and David JC MacKay. "Bayesian online changepoint detection." arXiv:0710.3742. 2007.

---

### Decision · Program_Chairs · 2019-12-19

**Decision:**

Reject

**Comment:**

In this paper the authors view meta-learning under a general, less studied viewpoint, which does not make the typical assumption that task segmentation is provided. In this context, change-point analysis is used as a tool to complement meta-learning in this expanded domain.

The expansion of meta-learning in this more general and often more practical context is significant and the paper is generally well written. However, considering this particular (non)segmentation setting is not an entirely novel idea; for example the reviewers have already pointed out [1] (which the authors agreed to discuss), but also [2] is another relevant work. The authors are highly encouraged to incorporate results, or at least a discussion, with respect to at least [2]. It seems likely that inferring boundaries could be more powerful, but it is important to better motivate this for a final paper.

Moreover, the paper could be strengthened by significantly expanding the discussion about practical usefulness of the approach. R3 provides a suggestion towards this direction, that is, to explore the performance in a situation where task segmentation is truly unavailable.

[1] Rahaf et el. "Task-Free Continual Learning".
[2] Riemer et al. "Learning to learn without forgetting by maximizing transfer and minimizing interference".